# Possible Effects of Uremic Toxins p-Cresol, Indoxyl Sulfate, p-Cresyl Sulfate on the Development and Progression of Colon Cancer in Patients with Chronic Renal Failure

**DOI:** 10.3390/genes14061257

**Published:** 2023-06-13

**Authors:** Rossella Di Paola, Ananya De, Raafiah Izhar, Marianna Abate, Silvia Zappavigna, Anna Capasso, Alessandra F. Perna, Antonella La Russa, Giovambattista Capasso, Michele Caraglia, Mariadelina Simeoni

**Affiliations:** 1Department of Mental and Physical Health and Preventive Medicine, University of Campania “Luigi Vanvitelli”, 80138 Naples, Italy; rossella.dipaola@unicampania.it (R.D.P.); ananyade15011997@gmail.com (A.D.); izraafiah@gmail.com (R.I.); 2Department of Precision Medicine, University of Campania “Luigi Vanvitelli”, 80138 Naples, Italy; marianna.abate@unicampania.it (M.A.); silvia.zappavigna@unicampania.it (S.Z.); michele.caraglia@unicampania.it (M.C.); 3Department of Oncology, Livestrong Cancer Institutes, Dell Medical School, The University of Texas, Austin, TX 75063, USA; anna.capasso@austin.utexas.edu; 4Nephrology and Dialysis Unit, Department of Translational Medical Sciences, University of Campania “Luigi Vanvitelli”, 80131 Naples, Italy; alessandra.perna@unicampania.it; 5Department of Sperimental Medical and Surgical Sciences, Magna Graecia University, 88100 Catanzaro, Italy; lrantonella@yahoo.it; 6Biogem S.c.a.r.l. Research Institute, 83031 Ariano Irpino, Italy; gb.capasso@unicampania.it

**Keywords:** colon cancer, uremic toxins, chronic renal failure, p-cresol, p-cresyl sulfate, indoxyl sulfate, dysbiosis

## Abstract

Chronic kidney disease (CKD) induces several systemic effects, including the accumulation and production of uremic toxins responsible for the activation of various harmful processes. Gut dysbiosis has been widely described in CKD patients, even in the early stages of the disease. The abundant discharge of urea and other waste substances into the gut favors the selection of an altered intestinal microbiota in CKD patients. The prevalence of bacteria with fermentative activity leads to the release and accumulation in the gut and in the blood of several substances, such as p-Cresol (p-C), Indoxyl Sulfate (IS) and p-Cresyl Sulfate (p-CS). Since these metabolites are normally eliminated in the urine, they tend to accumulate in the blood of CKD patients proportionally to renal impairment. P-CS, IS and p-C play a fundamental role in the activation of various pro-tumorigenic processes, such as chronic systemic inflammation, the increase in the production of free radicals and immune dysfunction. An up to two-fold increase in the incidence of colon cancer development in CKD has been reported in several studies, although the pathogenic mechanisms explaining this compelling association have not yet been described. Based on our literature review, it appears likely the hypothesis of a role of p-C, IS and p-CS in colon cancer development and progression in CKD patients.

## 1. Introduction

The human gut hosts a complex and abundant aggregation of bacteria, collectively called gut microbiota [1]. Diet, lifestyle, several chronic diseases and the intake of drugs are recognized to influence the composition of the intestinal microbiota. Gut bacteria play a crucial role in both maintaining immune and metabolic homeostasis and protecting against pathogens. The altered gut microbiota composition, known as gut dysbiosis, has been associated with the pathogenesis of many diseases, including cancer and infections [2].

Besides the loss of other vital functions, in chronic kidney disease (CKD), there is a progressive and irreversible decline of the renal ability to eliminate toxic wastes. The discharge of these substances in the gut changes the intestinal microenvironment and favors the development of intestinal dysbiosis in CKD patients. This is mainly attributable to the expansion of aerobic bacterial strains with urease activity responsible for the conversion of urea into ammonia and the formation of three main uremic toxins: p-Cresol(p-C), p-Cresyl Sulfate(p-CS) and Indoxyl Sulfate(IS) [3,4,5]. Moreover, in CKD patients, these metabolites tend to accumulate in the blood due to their reduced renal elimination [3] (IS total, normal concentration: 0.5 mg/L IS total, uremic concentration: 44.5 mg/L p-CS total, normal concentration: 2.9 mg/L p-CS total, uremic concentration: 43.0 mg/L) [6].

The systemic gathering of these uremic toxins is even more marked in hemodialysis patients since their high degree of binding to plasma proteins further limits their removal by the dialysis treatment [6].

The increased plasma levels of p-C, p-CS and IS in CKD patients exert a variety of detrimental effects, including cell damage, chronic systemic and local inflammation, coagulation disorders, immunosuppression, and intestinal permeability alterations [6,7,8,9].

It has been shown that IS determines a significant increase in the levels of some pro-inflammatory cytokines [10] that favor the passage of pathogens from the gut into the circulation [11]. Moreover, the intestinal uremic toxins induce an increase of the reactive oxygen species (ROS), and this pro-oxidant activity is responsible for the inhibition of heme oxygenase-1 (HO-1) and superoxide dismutase (SOD-2), two enzymes both having a cytoprotective and antioxidant activity [10]. It has also been reported that p-CS promotes immune dysfunction by suppressing some crucial functions of both macrophages [12] and Th1-type cells [13]. All the above-listed processes are considered pro-oncogenic, and this indicates that the dysbiosis-derived uremic toxins might increase the cancer risk in CKD patients.

In addition, the uremic toxins of our interest seem to be implicated in the development and progression of different types of cancer (ex., melanoma, renal carcinoma, esophageal carcinoma, etc.) and, in some cases, have been recognized as potential biomarkers of disease. An up to two-fold increase in the incidence of colon cancer (CRC) has been reported in patients with severe CKD [14]. A study conducted on a population of patients on maintenance hemodialysis undergoing a screening colonoscopy showed a significantly higher prevalence of adenomatous polyps and advanced adenomas compared to the general population [7].

Considering the above-listed pro-oncogenic effects of p-CS, IS and p-C and their typical increase both in the intestine and in the blood of CKD patients, we decided to conduct a literature review aimed at discovering whether the hypothesis of a role of p-CS, p-C and IS in the determinism of a higher colon cancer risk in CKD patients.

## 2. Materials and Methods

### 2.1. Conduct of Review

All relevant scientific articles were included in the analysis using a structured and methodical search approach, conducted in accordance with the Preferred Reporting Items for Systematic Reviews and Meta-Analyses (PRISMA) criteria.

### 2.2. Search Strategy and Study Selection

We searched the available literature focusing on the activity of uremic toxins (p-C, IS and p-CS) at both the systemic and intestinal levels. Studies that examined this topic were identified by a computerized search of all English-language articles in the main electronic databases (Medline, PubMed, NIH, Cochrane, UpToDate, etc.). We performed a systematic search for quality full-text articles by combining the following Medical Subject Heading (MeSH) terms: “colon cancer and uremic toxins”, “systemic effects of uremic toxins”, “uremic toxin production and activity”, “uremic toxins and dysbiosis”, “Indoxyl Sulfate and inflammation”, “p-Cresol and cancer”, “p-Cresyl Sulfate and cancer”, “indoxyl sulfate and cancer”, “metabolism of uremic toxins”, “intestinal effects of uremic toxins”, “intestinal dysbiosis in patients with chronic renal failure”, “intestinal dysbiosis and colon cancer”. Three hundred and sixty-three references were initially retrieved. One hundred and five references were excluded because they were irrelevant to our topic. One hundred and two articles were discarded after full-text analysis because they contained no information on uremic toxin activity. Seventy-three articles were also discarded because they were not in English or because they were abstracts, manuscripts by unpublished authors, or repository manuscripts. Finally, 17 papers were included in our analysis (Figure 1).

## 3. Results

### 3.1. Intestinal Dysbiosis in CKD

It is known that CKD patients tend to develop intestinal dysbiosis, mainly due to the failure to eliminate urea and waste substances through the kidneys. These molecules are consequently re-directed to the intestine for their metabolization and excretion [3]. The elevated urea levels in the intestine increase urease-producing aerobic bacterial species, responsible for both conversion of urea into ammonia and the production of uremic toxins, such as p-C, p-CS and IS. The presence of ammonia in the gut is responsible for a pH increase in the intestinal lumen, which induces chronic inflammation and enhanced permeability of the intestinal barrier, with the consequent absorption of the above-mentioned uremic toxins in the systemic circulation [3,8]. Enterobacteriaceae are among the dominant bacterial families in subjects with CKD, and they express the urease enzymes responsible for the formation of p-C and indole, the latter being the precursor of IS [8]. Therefore, the increase of uremic toxins during CKD is determined by both their exalted production by the dysbiotic gut microbiota and their reduced renal elimination [15].

### 3.2. Metabolism of p-Cresol, p-Cresyl Sulfate and Indoxyl Sulfate

IS, p-C and p-CS are uremic toxins directly or indirectly generated by specific intestinal bacteria and their circulating and urinary levels are increased in patients with CKD. In particular, IS is produced by the liver through reactions of sulphation and hydroxylation of the indole coming from the entero-hepatic circulation. Moreover, it is generated by the metabolic activity of tryptophan, mainly exerted by *E. coli*. IS is freely filtered into the glomerulus and excreted in the urine under normal conditions, while in subjects with CKD, its renal clearance is reduced and accumulates in the blood. This effect is additionally enhanced by its increased upstream production due to the intestinal dysbiosis associated with CKD. The high levels of circulating IS can induce a series of harmful effects, including an increase of oxidative stress, an enhanced permeability of the intestinal epithelium and an endothelium disruption that are paralleled by the increased expression of some inflammatory genes [16].

The other two uremic toxins, p-C and p-CS, are instead metabolites of tyrosine and phenylalanine, and they are directly produced by the dysbiotic intestinal bacteria. However, p-C undergoes bacterial transamination and decarboxylation to produce p-CS [9]. Even p-C and p-CS blood levels tend to increase in CKD patients due to both the increased intestinal production and the reduced renal elimination, with a series of deleterious effects. In particular, p-C induces vascular damage [17] and has a genotoxic effect on enterocytes [18]; while p-CS is involved in renal damage progression, the increase of ROS production, and the immunosuppression [10] (Figure 2).

### 3.3. Uremic Toxins and Cancers

p-CS toxin seems to be directly involved in the development and progression of some solid tumors. In fact, it has been shown that patients affected by CKD, with higher serum p-CS levels, have a greater risk of developing bladder cancer. p-CS induces an increase in intracellular ROS which determines the activation of the epithelium-mesenchymal transition (EMT), which is essential for tumor growth and migration [19]. Moreover, p-CS appears to be a regulator of the proliferation and migration of clear-cell renal cell carcinoma (ccRCC). It has been shown that p-CS induces the overexpression of the inducible hypoxia factor (HIF-1α) that is essential for the neo-angiogenesis process and regulates the expression of proteins related to EMT, including E-cadherin, fibronectin and vimentin. Furthermore, p-CS directly increases the expression of miR-21, one of the most frequently upregulated oncogenic miRNAs in solid tumors. Taken together, the present results demonstrate that p-CS directly induces the proliferation and migration of ccRCC cells through mechanisms involving the miR-21/HIF-1α signaling pathways [20]. Moreover, miR-21 was also reported to have an important role in the progression of colorectal cancer and was proposed as a fecal biomarker for the early diagnosis of the disease [21].

The uremic toxin p-C appears to have a pro-oncogenic role in bladder cancer by promoting the invasion and the migration of urothelial cells by increasing the expression of the matrix metallopeptidase 9 (MMP9) and the levels of RAS proteins, Ras homolog family member A (Rho A), the mechanistic phosphorylated target of rapamycin (p-mTOR) and protein complex functioning as a transcription factor (NF-κB). Notably, MMP9 is involved in the neo-angiogenetic process in different types of cancer [22].

Several reports indicate that some uremic toxins, such as p-CS and IS, could be used as potential cancer biomarkers. It was observed an increase of p-CS plasma levels in mice carrying xenografts of human melanoma vs. controls [23].

An increase in urinary IS concentration was observed in patients with melanoma. Interestingly, among several up-regulated compounds, IS showed the highest correlation coefficient with metastatic, which implies the hypothesis of the role of IS in cancer progression [24]. Elevated levels of IS have also been found in the urine of both mouse models and patients with gastric and cervical cancer, indicating the interplay of this toxin in a wide span of cancers [25,26]. In a recent metabolomics study performed in 88 esophageal squamous cell cancer (ESCC) patients and 52 healthy controls, Chen et al. have demonstrated that IS is one of the 6 circulating metabolites showing a higher diagnostic accuracy (0.885) in detecting ESCC [27]. In another metabolomics study conducted on renal cell cancer (RCC) patients, the levels of IS in kidney cancer tissue were associated with reduced compensatory renal cell growth with unfavorable effects on both renal function and clinical outcome in RCC patients [28]. In a study performed on the urine volatilome of 75 clear cell RCC patients compared to 75 healthy volunteers, p-C was identified as a powerful diagnostic marker with an accuracy of 81% in a six-biomarker panel [29]. Moreover, p-C was previously identified as a pre-nephrectomy diagnostic marker in a cohort of 23 patients (14 clear cell RCC and nine papillary RCC) and 23 healthy donors. It showed an about three-fold increase in the urine of RCC patients (p = 0.012) without any evidence of a prognostic role [30]. It was also demonstrated that the alteration of gut microbiota in breast cancer could alter the production of some metabolites such as p-C, succinate and cadaverine. Therefore, their increased plasma levels can serve as useful diagnostic markers in breast cancer [31].

Chiao-Yin Sun et al. has shown that the uremic toxins p-CS and IS promote the expression and activity of DNMT (DNA methyltransferase) in the kidney. DNMT is responsible for the hypermethylation of the Klotho gene, with a suppressing effect on its anti-aging renoprotective activity [32]. In this regard, several studies in the literature show that the increase in DNMT expression, and consequently the increase in DNA methylation, is associated with the development and progression of diseases such as cancer and autoimmune diseases [33,34]. As Klotho is recognized as a tumor suppressor gene [35,36], its inactivation by DNMT is a critical pathological mechanism that results in an increased likelihood of neoplasms development [37,38]. Therefore, the downregulation of Klotho mediated by p-CS and IS through the DNMT pathway might represent the epigenetic modification explaining the increased risk of cancer in CKD patients.

Furthermore, it has been reported that the uremic toxins of our interest, in particular p-CS and IS, appear to be responsible for the up-regulation of the activity of nuclear factor-kB (Nf-kB), which determines, in turn, the triggering of a series of pro-inflammatory processes and the increase in ROS production [38,39,40,41,42,43,44,45,46,47,48,49]. Nf-kB plays a key role in regulating the immune response to infections, and its dysfunctions have been linked to several types of cancer, inflammatory processes, and autoimmune diseases. It has been observed that this factor plays a crucial role in the development of colon cancer since it facilitates oncogenesis and metastasis by controlling the cell cycle and gene expression of apoptosis [50,51,52,53].

Uremic toxins can also exert a negative effect on the pharmacokinetics and/or pharmacodynamics of cytotoxic drugs used in cancer treatment. An example is the effect of IS and hippuric acid on the hepatic cell uptake of SN-38, the active metabolite of capecitabine. This can induce a reduction of about 40% of the plasma clearance of SN-38 with a consequent increase in the risk of drug-related side toxic effects [54]. Specifically, a hampering of SN-38 transportation carriers by IS and hippuric acid has been reported [55]. Moreover, IS and hippuric acid can have detrimental effects during statin administration, increasing the risk of rhabdomyolysis. This cytotoxic effect has been demonstrated in rhabdomyosarcoma cells concomitantly exposed to statins, IS and hippuric acid. The same did not occur when uremic toxins were given with cisplatin, suggesting no potentiating effects on one of the most used anti-cancer agents in rhabdomyosarcoma [55].

Another important action of IS is the regulatory activity of osteoclasts differentiation in degenerative bone diseases and in bone cancer and metastases. In fact, IS induces osteoclasts differentiation and bone reabsorption by binding with the Aryl hydrocarbon receptor (AhR) [56]. On the other hand, it was demonstrated that IS is able to inhibit breast cancer cells both in vitro and in vivo through the binding with AhR and pregnane X receptor (PXR), while the enhanced expression of IS-producing enzymes is a favorable survival marker of breast cancer patients [57]. Based on the cytotoxic ability of uremic toxins, several strategies for their targeted delivery in cancer cells have been developed. Ruthenium (II) complexes encapsulating p-C were synthesized and tested against different cancer cell lines showing apoptotic effects and the activation of endoplasmic reticulum stress [58].

As for IS, it has also been reported that it could be responsible, at least in part, for the thromboembolism associated with colon cancer, as demonstrated in the inferior vena cava of animals xenografted with colon cancer cells. In fact, metabolomics studies conducted on these animals revealed high plasma levels of IS, increased nuclear expression of AhR and an over-expression of tissue factor (TF) and plasminogen activator inhibitor 1 (PAI-1) in endothelial cells of vena cava, suggesting a role of IS in the activation of this thrombogenic pathway [59].

Another interesting concept was analyzed by O. Fourdinier et al., who observed that in patients with CKD, there is a decrease in the expression of mir-126 in correlation with the increase in the concentration of free IS [60,61]. Reduced levels of mir-126 expression have also been observed in colorectal cancer. It has been seen that the increase in the expression of mir-126 inhibits the proliferation and growth of colon cancer cells [62]. This could mean that the downregulation of mir-126, associated with the increase in free IS, might result in an enhanced risk of colon cancer development and progression in CKD patients.

### 3.4. Association of CKD Dysbiosis and Colorectal Cancer: In It a Concrete Hypothesis?

Gut dysbiosis has been described even in patients with colorectal cancer (CRC), and it has been hypothesized that the chronic intestinal inflammation associated with the microbiota alteration increases cancer growth and progression [62].

The analysis of different studies revealed that CKD and CRC patients show a similar gut microbiota composition. In both CKD and CRC, a significant increase in aerobic bacteria, especially *Enterobacteriaceae*, is recorded [8,63]. This bacteria family includes many pathogenic strains that are responsible for the development of chronic intestinal inflammation by the production of several molecules, such as p-C [8,64]. In particular, the intestinal content of *E. coli* is significantly increased in both diseases, with evidence of increased production of p-CS in CKD and the downregulation of DNA mismatch repair proteins in CRC [8,63,65]. Moreover, in all stages of CKD (KDIGO classification 2021: https://kdigo.org/wp-content/uploads/2017/02/KDIGO-Glomerular-Diseases-Guideline-2021-English.pdf, accessed on 15 October 2021) the increased plasma levels of p-CS have been correlated to the enhanced fecal expression of *Ruminococcus* [8]. Suggestively, a prevalence of *Ruminococcus* has also been found in CRC patients [65].

*Bacteroides* are also significantly increased in both CKD and CRC. They induce aromatic amino acid fermentation into potentially bioactive products, including phenols, indoles, and p-C, and exert carcinogenic effects through DNA alkylation and mutation [61,63].

In both diseases, *Lactobacillaceae* and *Bifidobacteriaceae* are significantly decreased. These two bacterial families are probiotic producers and play a key role in maintaining gastrointestinal homeostasis [8,64,66,67]. *Lactobacillaceae* prevent cancer development and dissemination [7,63]. *Bifidobacteriaceae* significantly contribute to *de novo* intestinal folate biosynthesis, whose deficiency can cause chromosomal instability and an increased risk of aneuploidy associated with rectal neoplasms [64] (Table 1). Confirming that the dysbiosis in CKD and CRC is comparable, in a study conducted on mouse models with colon cancer, a significant increase in uremic toxins (IS and p-CS) was observed, caused by the increase of their producer bacterial strains [68].

The alteration of the intestinal microbiota has a profound influence on immune responses and, consequently, on the chronic inflammatory processes since the presence of a greater number of pathogenic bacteria determines the stimulation of the local immune response, causing continuous low-grade inflammation [63]. Based on the described gut microbiota similarities between CKD and CRC patients, it is possible that the shared risk factor of CRC development and progression is intestinal dysbiosis.

### 3.5. Chronic Intestinal Inflammation Induced by IS

It is known that one of the key risk factors for the onset of CRC is chronic intestinal inflammation. In this regard, it has been reported that the uremic toxin IS, expressed at high blood concentrations in CKD patients, induces the chronic inflammation of intestinal epithelia and tight junctions. It has been observed that in mouse models with CRC, there is an increase in IS and p-CS [68].

In mice infused with IS, severe infiltration of lymphocytes in the lamina propria of the villi was observed. Blunt and fused villi and mild intestinal lymphangiectasia were also evident. Furthermore, it was observed that IS significantly increases the expression of two enzymes involved in inflammatory reactions in the intestine, namely cyclo-oxygenase-2 (COX-2) and inducible nitric oxidase (iNOS). Serum levels of pro-inflammatory cytokines were also evaluated following IS treatment, and significantly increased levels of tumor necrosis factor-α (TNF-α), interleukins IL-1β and IL-6 were found [10]. This evidence was also confirmed in another study also showing an increased release of nitric oxide (NO), besides the over-expression of nitric oxide synthase (iNOS) and cyclooxygenase-2 (COX-2). Furthermore, pre-treatment with IS was also associated with increased production of the pro-inflammatory cytokines TNF-α and IL-6 [69].

### 3.6. Immune Dysfunction Induced by p-CS and IS

Uremia-related immune dysfunction is the result of a complex interaction between the innate and adaptive immune systems with fine-tuning of immune activation and suppression pathways. Both immune responses are responsible for the onset of a series of pathological conditions that can favor neoplasms development and progression [70].

The uremic toxin p-CS participates in immune dysfunction in CKD patients. In mice with reduced renal function, it was found that p-CS significantly reduces peripheral B lymphocytes. In detail, p-CS suppresses the IL-7-induced STAT5 signaling, leading to the inhibition of CD43+ B cell proliferation [71]. Other studies also report an interference of p-CS on the release of IL-12 with inhibition of the anti-angiogenetic pathway sustained by the differentiation of naïve T cells into Th1 cells. On the other hand, p-CS increases the production of IL-10, which, in turn, inhibits the activity of macrophages through negative feedback. These data indicate that p-CS suppresses some macrophage functions, which contribute to host defense and may play a role in renal failure-related immune dysfunction [12]. In another study, it was confirmed that p-CS determines the suppression of type Th1 cell immune responses and suppresses the production of interferon-ɣ (INF-γ) [13]. Levels of end-stage differentiated CD8 + T cells are also positively correlated with the level of p-CS, indicating a premature aging phenotype of the immune system in CKD patients [72]. More recently, Borges Bonan and colleagues demonstrated that a combination of uremic toxins, including p-CS and IS, contribute to the increase of intermediate pro-inflammatory monocytes (CD14+ and CD16+) in patients with CKD [73].

IS also enhances the ability of macrophages to release ROS, contributing to chronic inflammation, tissue damage and neoplastic transformation either locally in the gut or systemically through its circulation to other sites [69].

### 3.7. Alteration of the Permeability of the Intestinal Epithelium Induced by IS

Gut dysfunction is characterized by an increase in the permeability of the intestinal epithelium, which is an indicator of a compromised mucosal barrier. The latter is formed by a layer of intestinal epithelial cells (IEC) joined by tight junctions. This structure prevents the crossing of the barrier by pathogens that are unable to reach the systemic circulation and allows only the selective reabsorption of molecules from the intestine. Any damage induced to IEC cells can lead to a series of harmful effects, including the uptake of toxic substances in the systemic circulation, which induces the activation of innate immunity, further damaging the intestinal epithelium [65]. The establishment of this pathological condition, responsible for greater exposure to pathogens and consequently to a chronic inflammatory process, determines a greater possibility of developing intestinal tumors.

Intestinal dysfunction emerges as one of the most common complications of CKD [8]. The trigger is represented by the increased metabolism of urea by resident bacteria with a great release of ammonia in the intestinal lumen. This induces detrimental environmental changes, which cause enterocyte damage and the alteration of intestinal permeability [3,8]. In CKD patients, the damage of the intestinal barrier also increases in response to the release of uremic toxins. It has been reported that IS significantly contributes to intestinal dysfunction through its interference with the expression of genes related to the intestinal epithelial junction. An increase in macroscopic intestinal damage (necrosis of the villi, edema, and ulceration), permeability and serum inflammatory factors were observed in mouse models treated with IS, confirming the implication of this toxin in IEC damage [11]. In vitro, experiments have shown that IS damages the intestinal barrier by inhibiting the mitophagic flow of IEC. The mitophagic process favors the elimination of damaged mitochondria and is essential for maintaining cellular homeostasis. In IEC cells of mice treated with IS, an increase in the mitochondria encapsulated in the autophagosome has been observed. Furthermore, failure to eliminate damaged mitochondria promotes the production of ROS, which significantly induces the amplification of both cell damage and inflammatory processes. The role of IS in the inhibition of mitophagic flow is determined by the inhibition of DRP1 activity (essential in mitochondrial fission) induced by IRF1 up-regulation, which, in turn, directly binds the DRP1 promoter region and inhibits its expression [11]. It was confirmed that the treatment with IS causes a drastic reduction of both the viability of enterocytes (Caco-2) and the transepithelial electrical resistance (TEER), which results in greater permeability of the intestinal epithelium [74].

### 3.8. Increase in ROS Production Induced by IS

Oxidative stress is a pathological condition caused by the disruption of the physiological balance between the production and elimination of ROS [75]. It is known that the excessive production of ROS engages in the onset of several types of cancers, including CRC. ROS activity can determine chronic inflammation and DNA mutations promoting several oncogenic phenotypes [75].

IS and p-CS induce oxidative stress in subjects with CKD. In detail, it has been found that p-CS increases oxidative stress in leukocytes, contributing to vascular damage, while IS is responsible for increased ROS production in IEC cells. The increase in oxidative stress induced by IS in the intestine is associated with intestinal epithelial dysfunction, as well as an increased chronic inflammatory status [10,11]. Furthermore, Adesso et al. showed that ROS production in IEC cell cultures was directly correlated to IS concentration in the suspension media [76]. In another study, it was shown that the pro-oxidant function of IS is determined by the inhibition of HO-1 and SOD-2 activity [10]. The pro-oxidant activity of IS was also demonstrated to act on the endothelium. Dou L. et al. studied the effects of IS on endothelial cells and reported a promoting activity on both ROS production and NADPH oxidase activity associated with interference with the production of the antioxidant glutathione [77]. Wei Ling Lau et al. conducted Histopathological Analysis studies on the colon of rats with renal failure, and besides an increase of pro-inflammatory molecules (COX-2, MCP-1, iNOS and gp91), found a significant reduction of both Nuclear factor erythroid 2-related factor 2 (Nrf2) and its target gene products (NQO1, catalase and CuZn SOD) [78]. Although their direct effect was not explored, a possible role in the downregulation of the Nrf2/keap1 system by the uremic toxins linked to gut dysbiosis in CKD is possible [79].

### 3.9. DNA Damage in Enterocytes and p-C

DNA damage is the main step for neoplastic transformation, and this is especially true for CRC development [80]. The gut environment can be altered by several factors favoring DNA damage in enterocytes. The increase of the local production and accumulation of dysbiosis-related uremic toxins seems to represent a major factor explaining the high incidence of CRC in CKD patients. In vitro experiments conducted by Al Hinai et al. demonstrated a dose-dependent ability of p-C to induce DNA damage in enterocytes [18]. The genotoxic activity of p-C was also confirmed in another study reporting a 10-fold increase in cell respiration, ATP synthesis, and DNA damage in enterocyte cultures assayed with even higher doses of p-C. In conclusion, p-C has been identified as a metabolic disruptor and a genotoxic factor in enterocytes [81].

## 4. Discussion

In recent years, intestinal dysbiosis has been widely described as a complication of CKD, with increasing evidence for several detrimental reflexes. Several uremic toxins explored in this review (p-C, IS, p-CS) derive from urea metabolism and are released in the intestine of all-stage CKD patients by their altered gut microbiota. The parallel tendency to the accumulation of these toxins in body fluids is proportional to the degree of renal function impairment. They also have a very high affinity for plasma proteins, and this explains their low clearance by dialyzer filters and the observation of the highest levels of these toxins in hemodialyzed patients. Based on our study of the literature, it appears that the increase of both intestinal and systemic concentrations of p-C, IS and p-CS is associated with the onset of a series of pathological processes that could promote the development of CRC in patients with CKD [3,4,5] (Table 2).

The increased incidence of several types of cancer, such as renal cell carcinoma, leukemia, and cervical and colorectal cancer in patients with CKD, is already known. Lees JS has recently reported that cancer risk in CKD patients is site-specific and accords with renal function decline. The possible determinants and pathogenetic processes have not been clarified and fully explored to date [82,83]. In our review, we focused on CRC and highlighted some of the possible causes that could explain its increased incidence in patients with CKD by analyzing the activity of three uremic toxins (p-C, IS and p-CS) that induce a number of potentially pro-oncogenic conditions. In this regard, in various reports, it was described that IS induces both intestinal chronic inflammatory processes through the promotion of COX-2 and iNOS enzymes [10], the increased permeability of intestinal epithelium, the inhibition of mitophagic flow [11], the increased intestinal ROS production, the inhibition of the activity of HO-1 and SOD-2, the increase of NAD(P)H oxidase activity and the decrease of glutathione levels [10,77]. Even p-CS appears to exert a role in promoting the oncogenesis process. This depends on a p-CS immunosuppressive activity linked to the interference on B lymphocytes and Th1-type lymphocytes proliferation and functioning [12,13,71].

Moreover, p-CS seems involved in the development and progression of bladder cancer and clear cell renal cell carcinoma. In bladder cancer, these effects were induced by the promotion of ROS production that, in turn, favors the activation of EMT [19]. In ccRCC, p-CS induces an increase in HIF-1α expression, EMT and mir-21 expression [20]. The up-regulation of mir-21 is particularly relevant since it plays a fundamental role in the progression of colon cancer and can be used as a fecal biomarker for early diagnosis of CRC [21]. Furthermore, p-CS and IS also seemed to be responsible for the up-regulation of the activity of the nuclear factor-kB (Nf-kB), which in turn determines the triggering of a series of pro-inflammatory processes and the increase in the production of ROS [34,35,36,37,38,39,40,41,42,43,44,45]. Finally, it has also been shown that -Cresol causes DNA damage in enterocytes in a dose-dependent manner [52,53] and is able to promote the invasion and migration of tumor cells by increasing MMP9 expression in bladder cancer [22] (Table 2).

Another interesting element, which could explain the increase in the incidence of CRC in subjects with CKD, is that the alteration of the microbiota, present in both diseases, is superimposable. In both illnesses, an increase of bacterial families responsible for the production of uremic toxins is recorded, another effect that could indicate a direct connection between the activity of toxins and CRC. Furthermore, this similarity could indicate that the dysbiosis that is generated in subjects with CKD, being comparable to that present in CRC, could reproduce the ideal environment for the development of CRC.

## 5. Conclusions

In conclusion, considering both the pro-oncogenic activities of p-CS, IS and p-Cresol carried out at the systemic level, and above all in the intestine, and the condition of dysbiosis typical of CKD patients which largely coincides with that present in patients with CRC, it is possible to hypothesize that such uremic toxins may be involved in the increase of the development of CRC in patients with CKD. Since this review is focused on a single aspect of a very complex issue, we suggest further studies based on cell cultures, in vivo experiments, and tumor microenvironment characterization to reveal more about colon cancer development in CKD patients.

## Figures and Tables

**Figure 1 genes-14-01257-f001:**
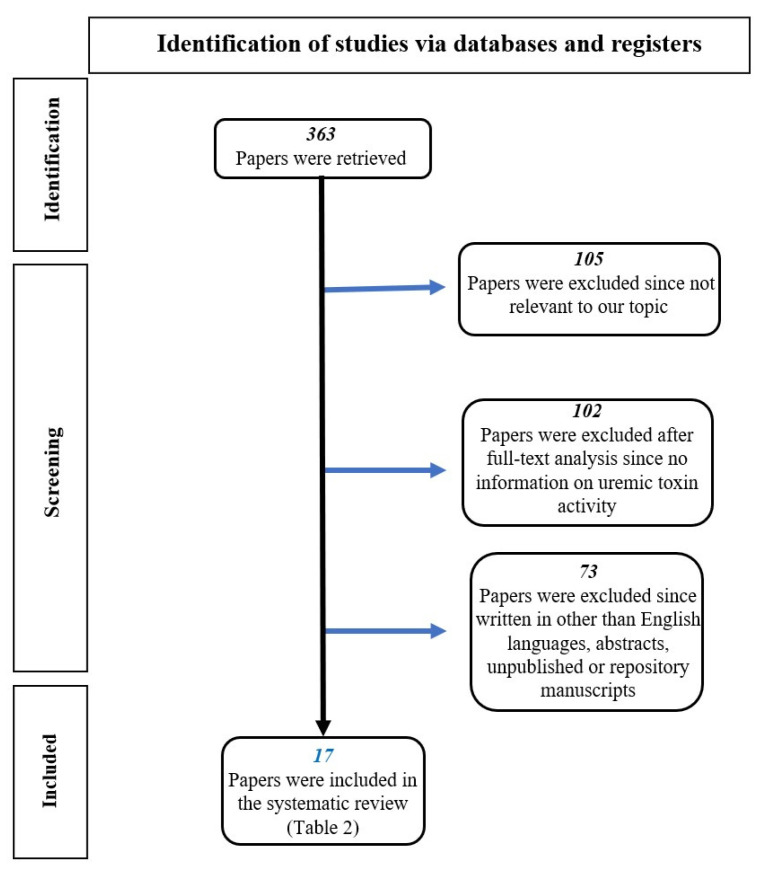
Flow chart of the literature selection process.

**Figure 2 genes-14-01257-f002:**
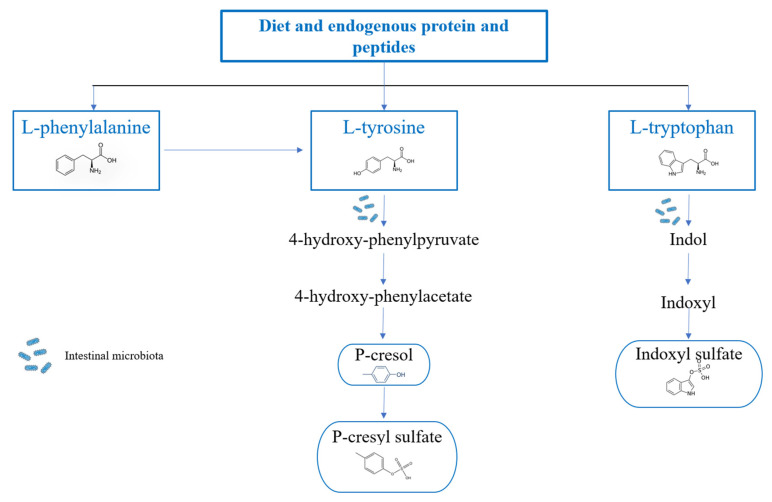
Route of conversion of tyrosine and phenylalanine to p-Cresyl Sulfate and of tryptophan to Indoxyl Sulfate. L-tyrosine is generally converted into 4-hydroxyphenylpyruvate by the enzyme tyrosine transaminase. 4-hydroxyphenylpyruvate is the precursor of 4-hydroxyphenylacetate, and this conversion occurs thanks to the activity of the enzyme p-hydroxyphenylpyruvate oxidase and can subsequently lead to the formation of p-Cresol by the p-hydroxyphenylacetate decarboxylase. In the intestinal mucosa and liver, most of the p-Cresol will be converted into the uremic toxin p-cresyl sulfate by aryl sulfotransferase. Phenylalanine, another aromatic amino acid, is also responsible for p-cresyl sulfate production through its conversion into tyrosine, mediated by phenylalanine 4-monooxygenase, also known as phenylalanine hydroxylase. The metabolization of tryptophan starts in the intestine, and the formation of indole occurs through the tryptophanase enzyme, which, once produced, is either metabolized by intestinal bacteria or released into the bloodstream up to the liver, where it undergoes hydroxylation, with the formation of indoxyl, and sulphation with the consequent formation of the final product.

**Table 1 genes-14-01257-t001:** Summary of the alterations of the intestinal microbiota of patients with CKD and CRC. Description of the activity of the different bacteria in patients with CKD and CRC. (+) = increase; (−) = decrease.

MICROBIOTA	CKD	CRC
+Enterobacteriaceae	Responsible to produce p-Cresol (p-C) [8]	Responsible for the development of chronic intestinal inflammation which favors tumor development [63]
+Escherichia coli	Mainly responsible for the formation of indoxyl sulfate (IS) [8]	-Promotes tumorigenesis -Down-regulates DNA mismatch repair proteins [64,65]
+Ruminococcus	Responsible for the production of p-Cresyl Sulfate (p-CS) [8]	Patients with a high risk of CRC have significantly higher levels of these bacteria [65]
+Bacteroides	Capable of fermenting aromatic amino acids to produce potentially bioactive products, including phenols, indoles and p-C [61]	Possess the ability to exert carcinogenic effects through the alkylation of DNA [63]
−Lactobacillaceae	Fundamental role in maintaining gastrointestinal homeostasis and preventing the development of cancer and the migration of cancer cells [8,64,66,67]	
−Bifidobacteriaceae	Contribute to de novo biosynthesis of folate in the intestine, and their deficiency can cause chromosomal instability and increased risk of aneuploidy associated with rectal cancer [64]	

**Table 2 genes-14-01257-t002:** Summary of the potentially pro-oncogenic effects of p-C, p-CS and IS.

Toxins	References	Role in Cancer	Study Design	Study Group
**p-CRESIL SULFATE** **(p-CS)**	Shiba T et al. [12]	↓macrophage activity	Experimental	Cells: RAW264.7 (macrophage-like cell line)
	Shiba T et al. [13]	↓Th1-type immune response	Experimental	Mouse model: female BALB/c mice cells: splenocyte cultures
	Peng YS et al. [19]	↑EMT ↑ROS	Experimental	Cells: TSGH-8301 (bladder cancer cells)
	Wu TK et al. [20]	↑EMT ↑HIF-1α ↑miR-21	Experimental	Cells: A498 and 786-O (human CRcc cells)
	Li F et al. [68]	↑IL-6	Experimental	Cells: Caco2 (normal colon cell line)
	Shiba T et al. [71]	↓peripheral B lymphocytes	Experimental	Mouse model with renal dysfunction
	Chiu YL et al. [72]	↑CD8+ T cells	Cross-sectional	412 patients with end-stage renal disease
	Bonan BN et al. [73]	↑CD14 ++ ↑CD16 +	Experimental	220 patients
**INDOXYL SULFATE** **(IS)**	Rapa SF et al. [10]	↑intestinal ROS production ↑SOD2 ↑HO1 ↑intestinal expression of COX 2 ↑iNOS ↑TNF-α ↑IL-6 ↑IL-1β	Experimental	Mouse model: C57BL/6J Cells: IEC-6(normal small intestine cell line)
	Huang Y et al. [11]	↑macroscopic intestinal damage ↑ pro-inflammatory cytokines ↑intestinal ROS production ↑IRF1 ↓DRP1	Experimental	Mouse models: IRF1 knockout, BALB/c with CKD Cells: Caco2 (normal colon cell line)
	Li F et al. [68]	↑IL-6	Experimental	Cells: Caco2 (normal colon cell line)
	Adesso S et al. [69]	↑(NO), ↑iNOS ↑COX-2 ↑TNF-α ↑IL-6	Experimental	Cells: macrophages J774A
	Bonan BN et al. [73]	↑CD14 ++ ↑CD16 +	Experimental	220 patients
	Tungsanga S et al. [74]	↓cells vitality ↓transepithelial electrical resistance	Experimental	Cells: Caco2 (normal colon cell line)
	Adesso S et al. [76]	↑intestinal ROS	Experimental	Cells: IEC-6(normal small intestine cell line)
	Dou L et al. [77]	↑ROS ↑NAD (P) H activity ↓Glutathione	Experimental	Cells: HUVEC (endothelial cell line)
**p-Cresol** **(p-C)**	Hsu YH et al. [22]	↑genotoxicity	Experimental	Cells: HT29 (colon cancer cell line) Caco2(normal colon cell line) IEC-6(normal small intestine cell line)
	Hinai EAA et al. [18] Andriamihaja M et al. [81]	↑genotoxicity ↓ATP	Experimental	Cells: HT29-Glc−/+ (colon cancer cell line) IEC-6(normal small intestine cell line)

## Data Availability

Not applicable.

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
