# Peer review of "Possible Effects of Uremic Toxins p-Cresol, Indoxyl Sulfate, p-Cresyl Sulfate on the Development and Progression of Colon Cancer in Patients with Chronic Renal Failure"

_genes, 2023, doi:10.3390/genes14061257_

Round 1
Reviewer 1 Report
In this manuscript, the author performed Possible effects of uremic toxins p-cresol,indoxyl sulfate,p-cresyl sulfate on the development and progression of colon cancer in patients with chronic renal failure.
Overall, this review is:
-comprehensive
- and well-organized.
Author Response
We are grateful to the reviewer for these positive comments.
Reviewer 2 Report
Several uremic toxins (p-C, IS, p-CS) identified in this review are derived from urea metabolism and are released in the gut of all patients with CKD via altered gut microbiome. The parallel tendency of these toxins to accumulate in body fluids is proportional to the degree of renal impairment. It appears that increased intestinal and systemic concentrations of p-C, IS, and p-CS are associated with the occurrence of a range of pathological processes in patients with CKD that may contribute to the development of colorectal cancer. But I have a question, is there an increased risk of developing other tumors in CKD patients? Patients with chronic kidney disease (CKD) develop intestinal flora dysregulation in the later stage, and many studies have found that various tumors are inextricably associated with the advocator flora dysregulation. Is it incomplete to explain the relationship between CKD and colon cancer solely from several urinary metabolic toxins? Please clarify these two questions.
Author Response
Several uremic toxins (p-C, IS, p-CS) identified in this review are derived from urea metabolism and are released in the gut of all patients with CKD via altered gut microbiome. The parallel tendency of these toxins to accumulate in body fluids is proportional to the degree of renal impairment. It appears that increased intestinal and systemic concentrations of p-C, IS, and p-CS are associated with the occurrence of a range of pathological processes in patients with CKD that may contribute to the development of colorectal cancer.
But I have a question, is there an increased risk of developing other tumors in CKD patients?
Answer: CKD patients are burdened by an increased oncologic risk as reported in several recent studies about colon cancer development in CKD patients. The most recent is by Lees JS on Nephrol Dial Transplant. 2023 May 4;38(5):1071-1079. We added this concept in the discussion (lines 438-445), listed other cancer types typically associated with CKD, and cited Lees JS paper. In the latter, it is stated that the increase of cancer risk in CKD patients is site-specific and accords with renal function decline. Moreover, the authors pointed out the importance of investigating the underlying molecular mechanisms.
Patients with chronic kidney disease (CKD) develop intestinal flora dysregulation in the later stage, and many studies have found that various tumors are inextricably associated with the advocator flora dysregulation. Is it incomplete to explain the relationship between CKD and colon cancer solely from several urinary metabolic toxins?
We thank the reviewer for these comments that surely will improve the quality of our work. We agree that the possible link between the CKD-related intestinal dysbiosis and colon cancer might not be the only mechanism explaining the increased risk of colon cancer in patients with renal failure. Instead we retain that intestinal dysbiosis is a complex process that can start even in early stages of CKD, as we reported in a recent study (reference n 68). Moreover, our manuscript describes an overlap in the gut bacterial populations that would link the two diseases. (section 3.4). In the current version, we added a sentence pointing out that we only focused on one aspect of a very complex issue and found important intestinal dysbiosis-related contacts between the two diseases. Moreover, we suggest further studies based on tumor microenvironment dissection for understanding more about colon cancer development in CKD patients. (lines 486-489)
Reviewer 3 Report
In this review, the authors describe the correlation of urinary levels of p-cresol, indoxyl sulfate, and p-cresyl sulfate with various phenomena in malignant tumors.
The focus is very intriguing, and the correlation with a wide variety of phenomena is concisely stated, so I think there is no problem in publishing it as it is.
In my personal opinion, the 17 references mentioned in Figure 1 should be listed as a single table.
Author Response
In this review, the authors describe the correlation of urinary levels of p-cresol, indoxyl sulfate, and p-cresyl sulfate with various phenomena in malignant tumors.
The focus is very intriguing, and the correlation with a wide variety of phenomena is concisely stated, so I think there is no problem in publishing it as it is.
In my personal opinion, the 17 references mentioned in Figure 1 should be listed as a single table.
Answer: We thank the reviewer for these positive comments and for the suggestion about the references report format. In the current version of the manuscript, we added the citation of Table 2 (where papers are listed and summarized) in Figure 1.